# Is a meditation retreat the better vacation? effect of retreats and vacations on fatigue, emotional well-being, and acting with awareness

Gerhard Blasche[1]*, Jessica deBloom[2,3], Adrienne Chang[4], Otto Pichlhoefer[5]

1 Department of Environmental Health, Center for Public Health, Medical University of Vienna, Vienna, Austria, 2 Department of Psychology, Faculty of Social Sciences, Tampere University, Tampere, Finland, 3 Faculty of Economics & Business, University of Groningen, Groningen, Netherlands, 4 Department of Sociology and Gerontology, Miami University, Miamí, Florida, United States of America, 5 Department of General Practice and Family Medicine, Center for Public Health, Medical University of Vienna, Vienna, Austria

* gerhard.blasche@meduniwien.ac.at

**Data Availability Statement:** The data file (SPSS 25) is provided in the supporting information.

**Funding:** The author(s) received no specific funding for this work.

## Abstract

It is well established that leisure vacations markedly improve well-being, but that these effects are only of short duration. The present study aimed to investigate whether vacation effects would be more lasting if individuals practiced meditation during the leisure episode. Meditation is known to improve well-being durably, among others, by enhancing the mental faculty of mindfulness. In this aim, leisure vacations during which individuals practiced meditation to some extent were compared with holidays not including any formal meditation practice as well as with meditation retreats (characterized by intense meditation practice) utilizing a naturalistic observational design. Fatigue, well-being, and mindfulness were assessed ten days before, ten days after, and ten weeks after the stays in a sample of 120 individuals accustomed to meditation practices. To account for differences in the experience of these stays, recovery experiences were additionally assessed. Ten days after the stay, there were no differences except for an increase in mindfulness for those practicing meditation. Ten weeks after the stay, meditation retreats and vacations including meditation were associated with greater increases in mindfulness, lower levels of fatigue, and higher levels of well-being than an "ordinary" vacation during which meditation was not practiced. The finding suggests that the inclusion of meditation practice during vacation could help alleviate vacations' greatest pitfall, namely the rapid decline of its positive effects.

## Introduction

It is well established that vacation, a several day respite from work, leads to a reduction of emotional exhaustion [1] and fatigue [2] both during vacation, as well as in the days thereafter. In addition, vacations can also improve well-being [3, 4], happiness, and life-satisfaction [5–7] and reduce perceived stress and negative mood [8–10]. A number of factors explain these

**Competing interests:** The authors have declared that no competing interests exist.

effects, including engaging in pleasurable activities [4], traveling to a holiday destination [7], higher ambient temperature and physical activity [11], the absence of negative incidences and negative work-related rumination [4, 12], natural environments [13] and having a respite from work [14]. Unfortunately, the positive effects of vacation on well-being decline rapidly after vacation, generally returning to pre-vacation levels within 1–3 weeks [1, 3, 15]. Returning to everyday life and/or work thus increases fatigue and decreases well-being. However, there are some exceptions to this rule. In a hiking vacation study, beneficial effects on well-being were still apparent seven weeks after the stay [10]. This suggests that activities known to improve health and well-being, such as physical activity, may lead to longer lasting and/or larger vacation effects. In line with this reasoning, recent studies found that both relaxation as well as physical activity conducted during rest-breaks prolonged their effect on well-being beyond usual respites from work [16, 17].

Apart from physical activity and relaxation, another activity known to improve well-being significantly is the practice of meditation [18]. Meditation can be understood as a family of diverse practices that include, but are not limited to, mindfulness meditation. Found within Buddhism and other contemplative traditions, such meditation practices seek to cultivate and regulate specific psychological processes, lastly aimed at increasing well-being. These practices have recently been grouped into three families, the attentional, the constructive, and the deconstructive family [19]. These families not only include different practices but also affect well-being through different avenues. The attentional family, encompassing mindfulness meditation and mindfulness-based stress reduction, improves well-being by a process known as "meta-awareness" or "cognitive distancing", i.e. "stepping back and observing one's internal processes of thinking and feeling" [19]. This requires overcoming experiential fusion, i.e. being absorbed in the contents of consciousness, thereby improving our ability to monitor and/or regulate psychological processes. To be able to promote these processes, two cognitive faculties have to be developed: *focused attention*, i.e. "sustaining selective attention moment by moment on a chosen object," and *open monitoring*, i.e. "nonreactively monitoring the content of experience" [20].

An extensively studied form of mindfulness meditation is mindfulness-based stress reduction (MBSR), a standardized meditation program. MBSR has been found to improve various facets of self-reported mindfulness [21] as well as well-being and perceived stress [22, 23], reduce emotional exhaustion [24] and improve mood [25]. For this type of meditation program, the improvement of well-being and mindfulness are correlated, indicating a close relationship between these two variables [23, 26]. Also, a dose-response relationship has been found in some [21, 23] but not all [27] studies, suggesting longer meditation practice to be associated with greater increases in mindfulness.

Meditation is frequently also practiced in the context of *meditation retreats*. Traditionally, meditation practices were developed within the context of one's spiritual path, with meditation retreats providing a period of intensive meditative practice for weeks, months, or even years, thus being a vehicle for a lifelong path of personal development and growth. In this way, meditation retreats differ from mindfulness programs, which are usually aimed at a clinical population with little prior meditation experience and characterized by a significantly smaller intensity of daily practice [28]. In a modern context, meditation retreats provide immersive, supportive environments where a practitioner may deepen his or her personal meditation practice in a sustained, continuous manner, removed from the obligations and distractions of daily life [28]. Motivations for participating in meditation retreats include gaining proficiency in particular forms of meditation, continuing one's ongoing meditative practice, progressing along one's personal spiritual path, seeking respite or recovery from stressful life events, or even participating in a form of leisure aimed towards personal development and self-

improvement [28, 29]. Residential meditation retreat programs, as the ones included in the present study, are often located in remote natural settings, where retreat participants receive instruction in meditation techniques and typically practice for one-week or longer. During such programs, participants may engage in multiple modalities of meditation for 8–9 hours a day, including formal sitting meditation practice, contemplations, and mindful movement exercises. In addition, participants have time for nature walks, writing, napping, reading, and simple kitchen or house chores.

The present study included meditation retreats embedded in a Buddhist tradition. The primary form of meditation taught at these retreats is Shamata (Sanskrit, 'calm abiding'), where the practitioners learn to focus their attention on an object of meditation, often the breath, as a form of support to stabilize the mind in present awareness. Through sustained practice, the meditator learns to stabilize the mind and develop greater awareness of the mind's plentiful activity: thoughts, emotions, sensations, perceptions. Shamata meditation encompasses both focused attention and open monitoring [20] and has been ascribed to the attentional family of meditation practices [19]. In its secular form, Shamata meditation can be regarded as a form of mindfulness meditation. However, while mindfulness practice taught during programs primarily aims at calming the mind, the goal of Shamata meditation is to familiarize oneself with one's inner world: to learn to direct and sustain one's attention in order to strengthen the capacity to be aware of one's myriad of mental processes: thinking, feeling, and perceiving and thus developing meta-awareness [19, 30].

Noted psychological benefits of meditation retreats include increases in trait mindfulness [31–35], decreased negative affectivity and increased perceived control [31], enhanced adaptive functioning [36], and an improvement of various facets of well-being [33–35, 37]. For example, one study compared a one-month Vipassana meditation retreat at a monastery with a control group pursuing life as usual, both groups consisting of experienced meditators [34]. During the retreat, individuals practiced meditation 8–9 hours per day. Variables were assessed directly at the beginning and end of the retreat. Results showed that the retreat led to an increase in self-reported mindfulness as well as several aspects of well-being such as positive and balanced affect. Also, the retreat affected some domains of personality including an increase in cooperativeness and a decrease in reward dependency.

Recently, two meta-analyses were conducted on the effects of meditation retreats. The first documented large improvements on measures of anxiety, depression and stress as well as on self-reported mindfulness and compassion both longitudinally as well as in comparison with a control group [38]. Effects declined but were still apparent at follow-up. Whereas effects were larger for novice meditators, no differences were found for different retreat types or for the duration of the retreat. However, there was a relationship between improvements in mindfulness and improvements in clinical outcomes. The second meta-analysis, focusing solely on mindfulness retreats, found that retreats outperform inactive controls conditions in improving mindfulness and well-being and that these improvements are maintained after the retreat [39]. All in all, the results imply that participants of retreats show improvements in self-rated mindfulness and well-being that persist for some time after the retreat. However, differences to *active* control conditions (e.g. stress management, vacation) remain inconclusive and warrant further research.

As stated above, meditation retreats are similar to vacations in some respects. Similarities include a respite from work or other obligations of everyday life, traveling to and staying at another place, and engaging in potentially relaxing activities that may provide new perspectives to one's life. Due to these common facets and keeping in mind the well-documented short term effect of vacation on health and well-being [1, 3], it is reasonable to expect retreats to have similar short-term effects as vacations regarding the improvement of well-being.

However, there are obvious differences between retreats and vacations as well. Whereas meditation retreats aim at improving one's meditation practice and progressing on one's spiritual path in a supportive environment, thus emphasizing the development of a skill, vacations provide leisure time to pursue preferred activities, including pleasurable and low effort activities [28, 40, 41]. Thus, retreats and vacations presumably differ regarding the extent of mastery and/or challenge that the individual experiences, with long hours of daily meditation undoubtedly being more challenging than, for example, lying on the beach, even though present-day vacations can also involve education, skills development, and self-improvement [29]. A second difference between retreats and vacations is that during vacations, the activities predominantly are under the individual's control and thus self-determined [42]. In contrast, activities during retreats, including meditation times and practices, mealtimes, and other activities, are generally predetermined. Thus, the extent to which an individual has control over his or her activities will differ.

It is known that the experience of a leisure episode can affect their impact on well-being [43, 44]. For example, mastery is negatively associated with post-leisure fatigue and positively with post-leisure vigor, suggesting that leisure episodes that are challenging lead to a decrease in fatigue and an increase in vigor. Though the experience of control has not been consistently found to affect fatigue, it is positively related to vigor. Two additional established leisure experiences, namely relaxation and detachment from work during leisure time, are associated with improved well-being [43].

The present study aimed to determine whether the rapid decline of well-being following a regular vacation could be attenuated by practicing meditation *during* the leisure episode as an activity known to improve well-being by various psychological mechanisms, including an increase in mindfulness. In this aim, we compared meditation retreats with individually planned vacations regarding their effect on mindfulness, fatigue, and emotional well-being in experienced meditators using a naturalistic observational design. To be able to account for the effect of meditation practice during the vacations, we distinguished between vacations during which the vacationers practiced meditation (vacations-with-meditation) and vacations during which vacationers did not practice any type of formal meditation (vacations-without-meditation). In this way, we could simultaneously compare the effect of different types of leisure episodes (retreats versus a vacation) while at the same time investigating the effect of meditation (retreats and vacation-with-meditation on the one hand compared to vacations-without-meditation on the other hand). In the present study, "acting with awareness" was chosen as a measure of mindfulness because it represents a core aspect of mindfulness [45, 46]. Therefore, we will use the term "acting with awareness" in the text when referring to the mindfulness measure used in this study.

Based on the reviewed effects of individual meditation and meditation retreats on self-reported mindfulness, we assume that meditation retreats, i.e., the intense, several day practice of meditation, will be associated with a higher level of acting with awareness after the leisure episode than vacation-with-meditation (hypothesis 1a) and that vacations-with-meditation will be associated with a higher level of acting with awareness than vacations-without-meditation (hypothesis 1b). Based on the reviewed effects of individual meditation and meditation retreats on well-being, we assume that meditation retreats also will be associated with a lower level of *fatigue* upon returning home than vacation-with-meditation (hypothesis 2a) and that vacations-with-meditation will be associated with a lower level of fatigue after the leisure episode than vacation-without-meditation (hypothesis 2b). Likewise, we assume that meditation retreats also will be associated with a higher level of *emotional well-being* after the episode than vacation-with-meditation (hypothesis 3a) and that vacation-with-meditation will be associated with a higher level of emotional well-being after the episode than vacation-without-meditation

(hypothesis 3b). Based on the known effect of meditation on mindfulness and well-being, we assume that the effects of meditation retreats and the effects of vacation-with-meditation, will be partly *mediated by an increase in* acting with awareness (hypothesis 4). Finally, based on the known effects of positive leisure experiences on well-being, we assume that the effects of both retreats and vacations are *partly mediated by the experience of these leisure episodes*, i.e., the experience of relaxation, mastery, and control (hypothesis 5).

## Material and methods

### Design

The study was a naturalistic observational study comparing three groups of individuals attending either one of several meditation retreats or an individually planned vacation during which individuals had or had not practiced meditation. The study was conducted with individuals of a large international Buddhist organization as the organization frequently offered meditation retreats. A convenience sample of five meditation retreats conducted from July to October was studied, three of which were in the US and two in Europe. These retreats were chosen for both size and content. The summer program season often attracts the highest participation of individuals. Additionally, all of these meditation retreats focused on intensive meditation practices, making them a relatively homogenous experimental setting. The type of meditation practice is outlined below. Individuals scheduled for these retreats were contacted via email, asking them to participate in a study on the effects of a meditation retreat on mindfulness and well-being as compared to a vacation (see S1 File). To constitute the vacation group, an email to all affiliates of the organization was sent by the organization asking affiliates whether they were planning a vacation in the specified period, i.e. July to October of the same year as the retreats, and if so, whether they were interested in participating in a study comparing meditations retreats with regular vacations (see S1 File). The provided vacation or retreat times were then used to trigger the invitation mails for the assessments with a link to an online questionnaire (Soscisurvey, Germany). Three assessments were made: assessment T1, the pre-stay assessment 10 days prior to the retreat or vacation, assessment T2, the post-stay assessment 10 days after the end of the retreat/vacation; and assessment T3, the follow-up assessment 10 weeks after the end of the retreat/vacation. The outcome variables fatigue, emotional well-being, and acting with awareness were assessed at all three-time points, individual characteristics at T1 and characteristics of the leisure episodes, including the experience thereof at T2. Individuals received a link to the questionnaire at the specified time and a reminder email two days later. To identify individual assessments, participants were asked to use an individualized but anonymous identification code for each assessment. The questionnaires were available in English and German. The complete text of the recruitment emails is provided in S1 File. The study was approved by the institutional review board of Miami University, Oxford, OH, USA 45056, with the project reference number: 01554e. Study participants gave informed consent by responding to the initial invitation letter. Data were analyzed anonymously.

### Study participants

Study participants were members or affiliates of a Buddhist organization scheduled to participate in one of five meditation retreats (retreat group) and members or affiliates of the same Buddhist organization planning a vacation. Specifically, 271 individuals planning to participate in one of the retreats and 181 individuals planning a vacation in the specified time, in sum 452 individuals, stated their willingness to participate in the study. Of these, a total of 321 individuals (71%) responded to the first questionnaire, 201 individuals (44%) also to the second and 129 individuals (28.5%) to all three questionnaires. Of the latter, three individual datasets had

to be removed due to missing data and nine due to stays shorter than seven days or longer than 36 days, leaving a total of 120 individuals in the study sample. The final sample did not differ in any of the descriptive or outcome variables from the respondents to the first wave (t-test, p>.17), indicating that dropouts were random. The characteristics of the study participants are presented in Table 1. A total of 76 women and 44 men participated in the study. Their mean age was 51.5 years (SD = 12.7); the youngest was 23 years, the oldest 81 years old. Individuals habitually practiced an average of 5 hours (SD = 5.3) of meditation per week at home, the average number of years since the beginning of meditation practice was 12.1 years (SD = 10.3), varying between 0 and 44 years. The English language questionnaire was used by 94 individuals, the German language questionnaire by 26.

## Description of retreat- and vacation stays

The study encompassed three meditation *retreats* of 2–5 week duration at meditation centers at three different rural locations (Barnet, Vermont, USA; Red Feather Lakes, Colorado, USA; and Aegina, Greece), including 7–8 hours of daily sitting meditation (n = 46); a 10-day Buddhist teaching program situated in a meditation center in rural France including an average of 5 hours of sitting meditation per day as well as oral teachings (n = 9); and other meditation programs at various locations (France, USA, Canada) including an average of 7 hours of meditation per day (n = 4). A typical retreat day begins at 7 am, with early morning sitting practice, followed by morning exercise and breakfast at 8 am. Teachings and formal sitting practice begin at 9 am and extend to the rest of the morning. Lunch includes silent mindful eating practices from the Japanese Zen tradition. The time from 1 pm-3 pm is reserved for work and

**Table 1. Characteristics of individuals and leisure stays.**

| | Meditation retreat | | Vacation-with-meditation | | Vacation-without-meditation | | total | | | |
| | (1) | | (2) | | (3) | | | | | |
| | m | sd | m | sd | m | sd | m | sd | p | group |
|---|---|---|---|---|---|---|---|---|---|---|
| n | 59 | | 37 | | 24 | | 120 | | | |
| Age (years) | 49.5 | 13.8 | 54.0 | 11.8 | 52.5 | 10.4 | 51.5 | 12.7 | 0.216 | |
| Sex (females) | 35 (59%) | | 24 (65%) | | 17 (71%) | | 76 (63%) | | 0.598 | |
| Partner (yes) | 20 (34%) | | 25 (68%) | | 14 (58%) | | 59 (49%) | | 0.003 | 1–2 |
| Paid work (yes) | 44 (75%) | | 29 (78%) | | 20 (83%) | | 93 (78%) | | 0.679 | |
| Language (English) | 55 (93%) | | 22 (60%) | | 17 (71%) | | 94 (78%) | | < .001 | 1–2 |
| Years of meditation practice | 10.1 | 10.2 | 14.1 | 11.0 | 13.9 | 8.9 | 12.1 | 10.3 | 0.108 | |
| Average habitual meditation practice per week (hours) | 5.3 | 7.9 | 4.9 | 6.6 | 4.2 | 7.1 | 5.0 | 7.3 | 0.801 | |
| Duration of the stay (days) | 13.8 | 7.9 | 15.4 | 8.1 | 15.0 | 6.5 | 14.5 | 7.7 | 0.554 | |
| Meditation practice per week during stay (hours) | 33.8 | 12.2 | 3.4 | 3.4 | 0.0 | 0.0 | 17.6 | 18.3 | < .001 | 1–2. 1–3. 2–3 |
| Relaxation (T2) | 12.5 | 4.0 | 15.9 | 3.2 | 15.3 | 4.1 | 14.1 | 4.1 | < .001 | 1–2. 1–3 |
| Mastery (T2) | 16.7 | 2.7 | 13.5 | 3.6 | 12.5 | 3.9 | 14.9 | 3.7 | < .001 | 1–2. 1–3 |
| Control (T2) | 10.2 | 3.8 | 14.9 | 3.6 | 14.6 | 3.8 | 12.5 | 4.3 | < .001 | 1–2. 1–3 |
| Acting with awareness (T1) | 29.7 | 5.7 | 31.9 | 5.8 | 30.4 | 6.1 | 30.5 | 5.9 | 0.198 | |
| Fatigue (T1) | 23.0 | 6.9 | 22.9 | 6.8 | 23.8 | 6.1 | 23.1 | 6.7 | 0.865 | |
| Emotional well-being (T1) | 19.1 | 4.8 | 18.8 | 5.0 | 18.3 | 4.2 | 18.8 | 4.7 | 0.766 | |

Note: T1: 10 days before the stay, T2: 10 days after the stay, T3: 10 weeks after the stay; significant (p < .05) group differences refer to differences between the 3 stays and are based on the Scheffé-Test indicated in the right hand column.

study, with formal sitting practice resuming from 3–6 pm, with periodic breaks. Evenings may include additional sitting practice or teachings. All retreats occurred in July to September.

The primary type of formal meditation taught at these retreats is *Shamata* meditation, belonging to the attentional family of meditation practices and involving focused attention and open monitoring [19]. An additional meditation practice engaged in on some of these meditation retreats was *Tonglen*, a traditional form of Tibetan Buddhist meditation that emphasizes loving-kindness and compassion. Tonglen can be classified in a family of constructive meditation practices in that it involves "systematically altering the content of thoughts and emotions," thereby leading to cognitive reappraisal [19]. In addition to formal meditation practice, individuals also were instructed to engage in *informal* meditation practices, such as cultivating and sustaining awareness while taking walks in nature, mindful eating, or doing household chores mindfully. These practices rely on focused attention and, therefore, can be viewed as belonging to the attentional forms of meditation practices [19].

Regarding the *vacations*, the most frequented destinations were the USA (16), Canada (8), France (6), Germany (5), and Spain (3). The vacations included in the study were leisure vacations. Vacation activities included spending time in rural or natural vacation locations (n = 32), traveling (n = 13), and visiting family (n = 5). Ten individuals did not specify their vacation activities. All vacations occurred from August to October. Some individuals indicated that they practiced meditation while on vacation on their own accord. Although we did not assess the type of meditation practiced, it is likely that individual meditation encompassed the practices described above, i.e., Shamata and Tonglen, as all individuals were members of the same Buddhist organization.

## Variables

*Acting with awareness* was assessed with one scale of the Kentucky Inventory of Mindfulness Skills (KIMS) (10 items, α = .86) [47, 48]. This facet of mindfulness was chosen as it represents a core component of mindfulness [45] and has a large impact on psychological well-being [26]. *Fatigue* was assessed with the Fatigue Assessment Scale (10 items, α = .86) [49]. Fatigue is a central variable in recovery from work and other sources of stress and thus potentially responsive for example to vacation [50–52]. *Emotional well-being* was assessed with the WHO-5 Well-being Index (5 items, α = .86) [53, 54]. Emotional well-being is a variable sensitive to stress as well as to recovery from stress and is also widely used in vacation research as well as research on mindfulness training [3, 46]. For the assessment of fatigue and emotional well-being, participants were asked to consider the time since the return from the retreat/vacation (at T2) or the last two weeks (at T1 and T3). All chosen questionnaires are widely used international scales with good scale properties. The experience of vacation was assessed with three scales of the Recovery Experience Questionnaire [44]. Relaxation (four items, e.g., "During retreat/vacation . . . I did relaxing things", α = .90), Control, (four items, e.g. "During retreat/vacation . . . I felt like I could decide for myself what to do", α = .88) and Mastery (4 items, e.g., "During retreat/ vacation . . . I learned new things", α = .83). The individuals were asked to "indicate to what extent the following statements describe your experience during the retreat or vacation."

## Data analysis

Data was analysed with Generalized Linear Model (SPSS 25) for three outcome variables at two points in time (T2 & T3) using two (acting with awareness) or three (fatigue and emotional well-being) models with a different number of predictor variables. The three outcome variables were acting with awareness, fatigue and emotional well-being at T2 and T3. In the first model, the following variables were entered: (a) the leisure episode (meditation retreats,

vacation-with-meditation, vacation-without-meditation) as categorial predictor variable, with vacation-with-meditation as reference category, (b) the respective dependent variable at T1 to be able to observe changes in outcomes, (c) age, having a partner, being engaged in paid work and language as control variables either because these variables differed significantly between groups (partner, language) or to control for other potential differences (age; paid work as engaging in paid work may increase fatigue and decrease well-being) [55]. In the second model, only calculated for fatigue and well-being, the residual of acting with awareness at T2 was entered to investigate its mediating effects. The residual acting with awareness (corrected for acting with awareness at T1) was used as an estimate of the *change* in this variable brought about by the leisure episodes. In the third model, relaxation, mastery and control were additionally entered into the model. To determine the differences between the groups (meditation retreats, vacation-with-meditation, vacation-without-meditation), simple analyses of variance or Chi$^2$ test were calculated. Effect sizes were calculated according to Cohen [56] using the estimated means plus the standard errors, which were transformed to standard deviations, derived by Generalized Linear Model (GLM) for model 1.

## Results

### Group and leisure episode differences, and intercorrelations

As expected, the three groups strongly differed regarding the hours of meditation practice during the leisure episodes (Table 1). During the meditation retreats, the average meditation practice was 33.8 hours per week and thus significantly longer than during vacation-with-meditation, where individuals meditated 3.4 hours per week. The group vacation-without-meditation included individuals who did not engage in meditation at all during their vacation. The leisure episodes also differed in regard to the recovery experiences essentially describing the individuals' experience of the particular leisure episode. Meditation retreats generally were perceived as providing more opportunities for mastery but fewer opportunities for control and relaxation compared to vacations in general. Vacations with and without meditation did not differ in any variable except for the practice of meditation.

The intercorrelation of variables is presented in Table 2. It should be noted that relaxation and control showed a fairly high correlation, indicating that a greater sense of control during the stay was associated with a greater experience of relaxation. As expected, the number of years of meditation practice was positively associated with acting with awareness. Furthermore, acting with awareness was negatively related to fatigue and positively with emotional well-being. Fatigue and emotional well-being showed a high negative correlation.

### Changes over time

To determine the change over time of the total study group, paired t-tests were calculated. Overall, the three leisure episodes taken together led to an increase in acting with awareness (t>5.5, p < .001) and emotional well-being (t>2.9, p < .004) and a decrease in fatigue (t>5.1, p < .001) both ten days (T2) and ten weeks (T3) after the episodes. Estimated means corrected for the four control variables are displayed in Fig 1.

### Differential effects of the three leisure episodes (hypotheses 1–3)

The results of the Generalized Linear Model (GLM) are presented in Tables 3–5. All analyses predicting acting with awareness, fatigue and well-being were significant for all models. To test hypotheses 1 (differences between leisure episodes in acting with awarness), hypothesis 2 (differences between leisure episodes in fatigue) and hypothesis 3 (differences between leisure

**Table 2. Intercorrelations of variables.**

|  |  | 1 | 2 | 3 | 5 | 6 | 7 | 8 | 9 | 10 | 11 | 12 | 13 | 14 |
|---|---|---|---|---|---|---|---|---|---|---|---|---|---|---|
| 1 | Age (years) |  |  |  |  |  |  |  |  |  |  |  |  |  |
| 2 | Sex (females) | .06 |  |  |  |  |  |  |  |  |  |  |  |  |
| 3 | Partner (yes) | .21 | .15 |  |  |  |  |  |  |  |  |  |  |  |
| 5 | Paid work (yes) | -.28 | -.00 | -.19 |  |  |  |  |  |  |  |  |  |  |
| 6 | Language (English) | .17 | .02 | .03 | -.09 |  |  |  |  |  |  |  |  |  |
| 7 | Years med experience | .41 | .11 | .27 | -.21 | .08 |  |  |  |  |  |  |  |  |
| 8 | Every-day practice (h/w) | .21 | -.01 | .14 | -.08 | .03 | .09 |  |  |  |  |  |  |  |
| 9 | Duration of the stay (days) | -.07 | .07 | .04 | -.01 | .04 | .21 | -.08 |  |  |  |  |  |  |
| 10 | Relaxation (T2) | -.09 | -.02 | .15 | -.01 | -.19 | -.05 | .03 | -.04 |  |  |  |  |  |
| 11 | Mastery (T2) | -.02 | -.03 | .06 | -.19 | .30 | -.03 | .21 | .01 | .09 |  |  |  |  |
| 12 | Control (T2) | .04 | .01 | .23 | -.01 | -.31 | .08 | .08 | .04 | .60 | -.05 |  |  |  |
| 13 | Acting with awareness (T1) | .23 | .20 | .11 | -.06 | -.14 | .32 | .25 | .00 | .07 | -.09 | .12 |  |  |
| 14 | Fatigue (T1) | -.26 | -.10 | -.05 | .24 | -.19 | -.21 | -.26 | .00 | -.04 | -.10 | -.11 | -.50 |  |
| 15 | Emotional well-being (T1) | .27 | .07 | -.03 | -.22 | .19 | .20 | .22 | .01 | -.01 | .09 | -.04 | .39 | -.75 |

Note: T1: 10 days before the stay, T2: 10 days after the stay; Coefficients r≥.18 are significant (p < .05).

episodes in emotional well-being), the type of leisure episode was entered into the analyses together with the corresponding outcome variable at T1 and the four control variables (Model 1). At T2 and T3, acting with awareness did not differ between vacation-with-meditation and meditation retreats (Table 3). Thus, hypothesis 1a was not confirmed. However, both at T2 and T3, acting with awareness was higher after vacations with meditation than after vacation-without-meditation, resembling a medium effect (d = .50), thus supporting hypothesis 1b. Pairwise comparisons revealed that acting with awareness was also higher following meditation retreats than following vacation-without-meditation both at T2 (p = .03, d = .51) and T3 (p < .001, d = .92). At T2, the leisure episodes did not differ in fatigue or emotional well-being (Tables 4 and 5). However, at T3, fatigue was lower (d = .62) and emotional well-being was higher (d = .70) after vacation-with-meditation than after vacation-without-meditation, whereas meditation retreats and vacation-with-meditation did not differ in fatigue or emotional well-being. Therefore, hypotheses 2a and 3a were not supported for T2 or T3 nor were hypotheses 2b and 3b supported for T2, but hypotheses 2b and 3b were supported for T3. In addition, pairwise comparisons revealed that fatigue was lower (d = .80) and emotional well-being higher (d = .81) after meditation retreats than after vacation-without-meditation at time point T3 (p < .001). To summarize, we found no differences between the leisure episodes at T2 in fatigue or emotional well-being, but higher levels of acting with awareness following leisure episodes including meditation. At T3, both vacation-with-meditation and meditation retreats outperformed vacation-without-meditation in all three variables to a medium to large degree, while meditation retreats and vacation-with-meditation showed similar effects.

## Moderating effects of acting with awareness (hypothesis 4)

To investigate the effect of the leisure period related change in acting with awareness, residual acting with awareness at T2 (corrected for acting with awareness at T1) was additionally entered into the model (Model 2). Residual acting with awareness was a significant predictor both of fatigue and emotional well-being at T2 and T3 (Tables 4 and 5). The inclusion of acting with awareness into the model reduced the overall impact of the leisure episodes on fatigue at T2 and T3 as can be seen in the reduction of leisure episode Wald Chi-Square (Table 4).

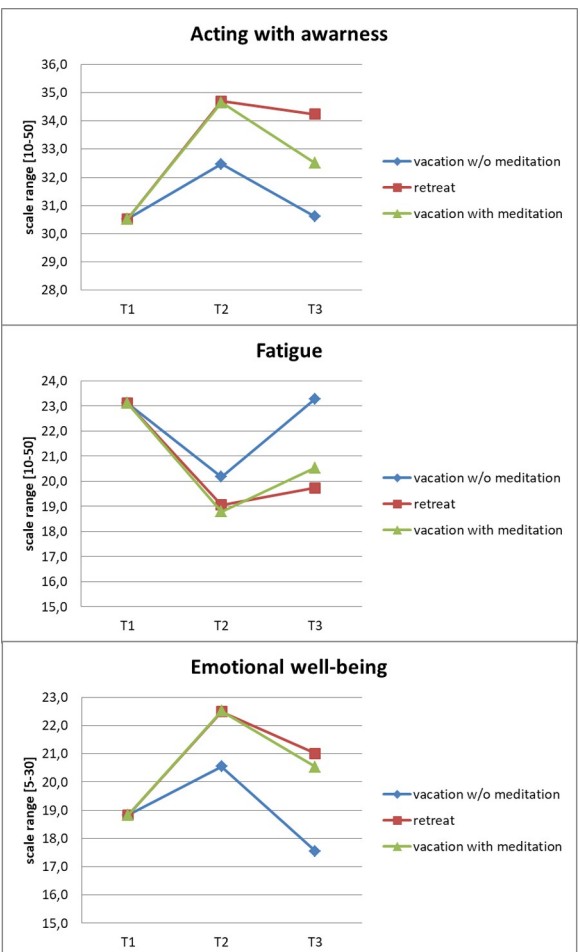

**Fig 1. Estimated means of outcome variables.** Note: Means corrected for the corresponding variable at T1, partner, age, language, paid work.

Similarly, the inclusion of acting with awareness reduced the impact of the leisure episodes on emotional well-being at both time points (Table 5). Taken together, these findings indicate that the change in acting with awareness partially mediated the effect of the leisure episodes on fatigue and well-being and thus supports hypothesis 4.

## Moderating effects of leisure experiences (hypothesis 5)

To investigate the effects of leisure experiences, the variables control, mastery and relaxation were additionally entered into the model (Model 3, Tables 3–5). We expected the experience of the leisure episode regarding control, mastery and relaxation to moderate the effect of the leisure episodes on the outcome variables. This hypothesis was partly confirmed for the experience of relaxation, which predicted fatigue and well-being at T2. Thus, hypothesis 5 was confirmed for relaxation at T2.

## Effects of experience with meditation

To test possible effects of the years of meditation experience on the stay related changes in acting with awareness, fatigue, and emotional well-being, we conducted six additional GLM

**Table 3. Results (GLM) predicting acting with awareness after vacation/retreat.**

|  | Acting with awareness at T2 | | Acting with awareness at T3 | |
| --- | --- | --- | --- | --- |
| Model | 1 | 3 | 1 | 3 |
| Acting with awareness (T1) | .55** | .55** | .50** | .48** |
| Meditation retreats | .04 | .98 | 1.72 | 3,09** |
| Vacation-with-meditation | ref | ref | ref | ref |
| Vacation-without-meditation | -2.18* | -1.93* | -1.90* | -1.87* |
| Partner (no<yes) | -.67 | -1.18 | -.30 | -.32 |
| Age (ys) | -.01 | .84 | .05 | .07* |
| Language (German<English) | .16 | .39 | -.30 | -.06 |
| Paid work (no<yes) | -.88 | -.47 | -.25 | -.11 |
| Control (T2) |  | .29** |  | .07 |
| Mastery (T2) |  | .18 |  | -.09 |
| Relaxation (T2) |  | .11 |  | .23* |
| Chi Square (leisure episode) | 5.8 | 6,7** | 15.3** | 19.0** |
| Chi Square (total model) | 65.4** | 85.5** | 73.6** | 82.0** |

Note: T1: 10 days bevor the stay, T2: 10 days after the stay, T3: 10 weeks after the stay; regression coefficients B are displayed;

**: p < .01,

*: p < .05

analyses for these three outcome variables at T2 and T3, with the same variables as in Model 1, plus the number of years of meditation practice as additional independent variable. For none of the analyses the effect of years of meditation practice was significant, with effects varying between a relatively high prediction for acting with awareness at T3 (B = -.05, p = .156) and a relatively low prediction for emotional well-being at T3 (B = -.0001, p = .983). This indicates

**Table 4. Results (GLM) predicting fatigue after vacation/retreat.**

|  | Fatigue at T2 | | | Fatigue at T3 | | |
| --- | --- | --- | --- | --- | --- | --- |
| Model | 1 | 2 | 3 | 1 | 2 | 3 |
| Fatigue (T1) | .52** | .52** | .52** | .60** | .60** | .59** |
| Meditation retreats | .27 | .30 | -.24 | -.80 | -.79 | -1.18 |
| Vacation-with-meditation | ref | ref | ref | ref | ref | ref |
| Vacation-without-meditation | 1.39 | .03 | -.02 | 2.74* | 2.11 | 2.05 |
| Partner (no<yes) | .61 | .20 | .26 | -.23 | -.42 | -.17 |
| Age (ys) | -.01 | -.01 | -.03 | -.04 | -.04 | -.06 |
| Language (German<English) | -.78 | -.70 | -.66 | .84 | .88 | .92 |
| Paid work (no<yes) | .81 | 1.35 | 1.46 | 1.59 | 1.84 | 2.01* |
| Residual acting with awareness (T2) |  | -.62** | -.58** |  | -.29** | -.22** |
| Control (T2) |  |  | .07 |  |  | .02 |
| Mastery (T2) |  |  | .00 |  |  | -.10 |
| Relaxation (T2) |  |  | -.24* |  |  | -.22 |
| Chi Square (leisure episode) | 1.7 | 0.2 | .06 | 11.5** | 7.8* | 6.7* |
| Chi Square (total model) | 65.2** | 114.5** | 12.7** | 85.5** | 94.9** | 101.0** |

Note: T1: 10 days bevor the stay, T2: 10 days after the stay, T3: 10 weeks after the stay; regression coefficients B are displayed;

**: p < .01,

*: p < .05

**Table 5. Results (GLM) predicting emotional well-being after vacation/retreat.**

| Model | Well-being at T2 | | | Well-being at T3 | | |
|---|---|---|---|---|---|---|
| | 1 | 2 | 3 | 1 | 2 | 3 |
| Emotional well-being (T1) | .42** | .45** | .43** | .54** | .56** | .55** |
| Meditation retreats | -.02 | -.03 | .23 | .47 | .48 | .76 |
| Vacation-with-meditation | ref | ref | ref | ref | ref | ref |
| Vacation-without-meditation | -1.96 | -.91 | -.77 | -2.98** | -2.35* | -2.28* |
| Partner (no<yes) | .24 | .59 | .23 | 1.16 | 1.36 | 1.17 |
| Age (ys) | .00 | .00 | .02 | .02 | .02 | .03 |
| Language (German<English) | 1.34 | 1.21 | 1.01 | -.47 | -.56 | -.62 |
| Paid work (no<yes) | -.38 | -.85 | -1.12 | -1.89* | -2.17* | -2.28* |
| Residual acting with awareness (T2) | | .47** | .38** | | .2** | .23* |
| Control (T2) | | | -.10 | | | -.02 |
| Mastery (T2) | | | .16 | | | .07 |
| Relaxation (T2) | | | .34** | | | .15 |
| Chi Square (leisure episode) | 4.3 | 1.1 | 1.0 | 12.3** | 8.3* | 6.8* |
| Chi Square (total model) | 33.5** | 61.1** | 77.4** | 49.7** | 58.9** | 61.8** |

Note: T1: 10 days bevor the stay, T2: 10 days after the stay, T3: 10 weeks after the stay; regression coefficients B are displayed;

**: p < .01,

*:p < .05

that the number of years of meditation practice did not affect the overall change during the leisure episodes.

## Discussion

The present study sought to investigate whether meditation practiced in the context of a leisure episode would promote (i.e., prolong) known beneficial effects on well-being. In this effort, the effect of two different types of leisure episodes, namely meditation retreats and vacations, on mindfulness, fatigue, and emotional well-being was studied in a sample of individuals with meditation experience. To be able to account for the effect of meditation, vacations during which individuals practice meditation at their own discretion (vacation-with-meditation) were distinguished from vacations during which meditation was not practiced (vacation-without-meditation). In addition, some mechanisms bringing about well-being improvement were investigated by determining the mediating effects of the change in mindfulness, as well as the mediating effect of the experience of the leisure episodes (i.e., relaxation, mastery and control). We assumed that meditation practice would increase mindfulness and in consequence foster the effect of the leisure episode on well-being. Thus, we expected meditation retreats to outperform vacations with meditation, and vacations with meditation to outperform vacations without meditation regarding both the increase in mindfulness, as well as the improvement in well-being. In addition, we expected leisure episodes experienced as relaxing, providing opportunities for mastery, and allowing control, to lead to a greater improvement in well-being.

The outcomes of the leisure episodes were assessed 10 days (T2) and 10 weeks (T3) after the end of the stays. Ten days after the leisure episodes, there were no differences between the three types of leisure episodes (i.e. meditation retreats, vacation-with-meditation and the vacation-without-meditation) regarding well-being variables, but meditation retreats and vacations with meditation lead to a greater increase in mindfulness than vacations without meditation. At first glance, this lack of differences in well-being is surprising and not in line with our hypotheses.

However, it is in line with the well-established short-term effects of vacation on well-being which was also found in the present study [1, 57], potentially masking the effects of meditation practice and the corresponding increase in mindfulness. These findings are also in line with research showing that individual differences in well-being markedly decline during vacation and in the first one or two weeks after vacation. For example, a vacation reduces rumination and affective well-being in obsessive versus non-obsessive workers [58]. In other words, just being away from work and/or everyday life and enjoying leisure makes most of us happy on a short term basis, independent of our personal characteristics. This may also apply to individual differences in mindfulness brought about by the practice of meditation during the leisure episodes. However, our results and the former reasoning are at variance with one study showing that trait mindfulness did affect the short-term improvement of exhaustion and vigor during a leisure weekend in a positive fashion [59]. Future studies will have to resolve this inconsistency.

Interestingly, ten weeks after the leisure episodes, acting with awareness and emotional well-being were higher and fatigue was lower following meditation retreats and vacation-with-meditation compared to vacation-without-meditation. This is in line with previous research both on the effects of mindfulness training [22, 27], as well as on the effects of meditation retreats [38, 39] showing that meditation practice and/or retreats not only improve mindfulness but also well-being in a durable fashion. The superior long-term outcome of meditation retreats and vacation-with-meditation (compared to vacations-without-meditations) on fatigue and emotional well-being are partly due to the greater increase in acting with awareness found for the leisure episodes during which meditation was practiced. As described above, 10 days after the leisure episode, it is likely that transient factors associated with the respite from work and everyday life brought about the increases in well-being, thereby masking the effect of mindfulness. At 10 weeks after the episode, when individuals had resumed work and/or their every-day chores, the more enduring practice-related improvements in acting with awareness most likely accounted for the sustained improvement of fatigue and emotional well-being found in those leisure episodes including meditation, considering the rapid post-vacation decline of well-being observed otherwise [3, 15]. Thus, meditation practice has the potential to make the effects of vacations more lasting.

As suggested above, leisure episode related changes in acting with awareness were at least partly responsible for related changes in fatigue and emotional well-being in the present study. Increases in acting with awareness were associated with a decrease in fatigue and an increase in emotional well-being both 10 days and 10 weeks after the leisure episodes. In addition, the fact that changes in acting with awareness explained some of the differences between the three leisure episodes regarding well-being and the fact that the leisure episodes differed in their effect on acting with awareness imply that acting with awareness partly moderates the effect of the leisure episodes on well-being. In other words, the increase in acting with awareness partly explained the effect of the leisure episodes on fatigue and emotional well-being. This result is in line with several studies finding practice related increases in mindfulness to be associated with increases in well-being both during meditation programs [23, 26, 60] as well as during retreats [38] and implies that one factor explaining the increased and prolonged effect of the leisure episodes in the present study is indeed an increase in mindfulness. It should be noted that other psychological processes associated with meditation practice not included in this study might also play a role in improving well-being, such as self-compassion [61, 62], neuroticism, and perceived control [31]. We suggest that future research on vacation, meditation retreats, and regular meditation practice should consider a broader range of potential mediators.

Contrary to our expectation, meditation retreats with several hours of meditation per day were not superior to vacations during which individuals practiced meditation for a few hours per week, neither regarding mindfulness nor in regard to well-being. This is at variance with

the assumption that more hours of meditation practice should bring about greater improvements of both mindfulness and well-being [21, 23]. However, there are some reasons why this may not be the case. Firstly, the meditation retreats differed in relevant aspects of leisure experience from the vacations in this study. Vacations were experienced as more relaxing and more under the individual's control than the meditation retreats. As both of these recovery experiences are associated with a greater decline in fatigue and/or a greater increase in vigor [43], the more beneficial experience of vacations may have compensated for less frequent meditation practice during vacation. Secondly, a meta-analysis on the impact of group-based mindfulness training on self-reported mindfulness did not find a dose-response relationship between the number of sessions and the increase in mindfulness [27]. Thus, the extent of meditation practice during a leisure episode may have less effect on mindfulness and well-being than whether one practices meditation during this leisure episode at all. This suggests that a limited amount of meditation practice during a vacation can sustain the positive vacation effects on well-being. However, other studies found that long-term retreat outcomes dependent on the extent of daily practice [28].

Despite this lack of differences between meditation retreats and vacations during which individuals practiced meditation, meditation retreats are a setting where individuals can learn the practice of meditation in the first place. It might also be the case that long-term meditators have acquired skills over their years of meditation practice that continued into post-meditation where they do not formally meditate, such as on vacation. Indeed, in the present study, the retreat participants had somewhat (though non-significantly) fewer years of meditation practice than the vacationers.

Next to the improvement of acting with awareness, the experience of the leisure episodes impacted fatigue and emotional well-being. Experiencing leisure episodes as relaxing was related to a greater decrease in fatigue and a greater increase in emotional well-being 10 days after returning from the vacation and/or retreat. This is in line with previous research finding the experience of relaxation during a vacation and/or a weekend to be related to greater post-vacation health and well-being [41], to less perceived effort while conducting work after vacation [12], and to more positive affective states following the weekend [63]. However, these effects were not found 10 weeks after vacation. Also, the other facets of leisure experience assessed in the present study, namely mastery, and control, were not related to the change of fatigue or emotional well-being after vacation.

Meditation practice, despite its overall beneficial effect on well-being, can also be associated with adverse events, such as anxiety, depression, or cognitive anomalies. The overall prevalence of these adverse events is 8.3% and thus, similar to that found for psychotherapy [64]. These unpleasant meditation-related experiences are more common in non-religious participants, participants with higher levels of repetitive negative thinking, and in those engaging in deconstructive types of meditation such as insight meditation [65].

Four potential limitations of the present study need to be addressed. Firstly, the participants were individuals with an average of 10 years of meditation practice and members or affiliates of a Buddhist organization. Apart from being more accustomed and more inclined to the practice of meditation than the general population, it is also likely that they exhibit higher levels of mindfulness [27]. Higher levels of trait mindfulness, as well as meditation practice per se, are associated with higher levels of well-being, as mindfulness has been shown to reduce emotional reactivity and psychological distress [22]. In addition, individuals with higher levels of trait mindfulness may benefit more from formal meditation practice [66] as well as leisure episodes [59]. Perhaps the long-term training in meditation creates healthy patterns of disengagement from stressful external stimuli (such as work) which compound rumination so that moderate practice of meditation would reinvigorate the acting with awareness. Thus, it is possible that

the individuals in the present study responded better both to the leisure episodes, as well as to the practice of meditation, potentially leading to greater improvements in acting with awareness, fatigue, and emotional well-being than in individuals not practicing meditation. However, an association of years of meditation practice with outcomes was not found in the present study. Thus, we conclude that the present study results can cautiously be generalized to the general population, although the general population may not be as inclined to practice meditation during leisure episodes to the same extent as the individuals of our study.

Secondly, we did not assess the extent to which the leisure episodes affected regular meditation practice in the aftermath of the stays. It cannot be ruled out that the lasting effect of retreats on mindfulness and well-being was due to an increase in everyday mindfulness practice as has been reported elsewhere [28]. However, as the average home-based meditation practice was fairly intense and did not differ between the groups before the leisure episodes, we do not believe that a substantial change in practice duration occurred after the retreat, though this cannot be ruled out. Future studies would have to assess the impact of meditation retreats on the extent of home-based meditation practice.

Thirdly, the study design was naturalistic observational. Individuals were not randomized to the various leisure episodes but had chosen these episodes on their own accord. This implies that the sample was selective and that the group comparisons we made should be regarded with caution, despite our efforts to control for group differences statistically.

Fourthly, we used a relatively short scale (i.e., 5 items) to assess emotional well-being. Even though this scale has good psychometric qualities demonstrating high levels of reliability, validity as well as sensitivity [67], future research could replicate our findings with more comprehensive scales and broaden the well-being construct to include eudaimonic well-being, thriving and meaning [68, 69].

## Conclusions

To conclude, we found that meditation retreats, as well as vacations during which vacationers practiced meditation at their own discretion, were associated with greater medium-term increases in mindfulness and emotional well-being than an "ordinary" vacation during which meditation was not practiced. Differences between meditation retreats with several hours of meditation per day and vacations with a few hours of meditation practice per week were negligible. In particular, 10 weeks after the leisure episodes including meditation, acting with awareness, and emotional well-being were higher and fatigue was lower than after "ordinary vacations" without meditation. From a theoretical point of view, this supports previous findings on the beneficial effects of mindfulness training on psychological well-being. From a practical point of view, the findings suggest that the inclusion of meditation during vacation could help to alleviate vacations' greatest pitfall, namely the rapid decline of its positive effects, thus making the benefits of vacation on fatigue and well-being more lasting.

Despite our effort to determine some mechanisms explaining the improvement of well-being, such as acting with awareness, future research should include a variety of theory-driven measures to extend our knowledge on the psychological mechanisms promoting well-being durably, including the assessment of the extent and type of post-retreat mediation practice [28], self-report measures such as acceptance, self-compassion, and empathy [19] as well as behavioral and physiological measures of attention [19, 28].

## Supporting information

**S1 File. Recruitment emails.**
(PDF)

**S2 File. SPSS data file containing all relevant data.**
(SAV)

## Author Contributions

**Conceptualization:** Gerhard Blasche, Otto Pichlhoefer.

**Data curation:** Adrienne Chang, Otto Pichlhoefer.

**Formal analysis:** Gerhard Blasche.

**Methodology:** Gerhard Blasche, Jessica deBloom.

**Project administration:** Adrienne Chang, Otto Pichlhoefer.

**Resources:** Adrienne Chang.

**Supervision:** Jessica deBloom, Otto Pichlhoefer.

**Validation:** Adrienne Chang.

**Writing – original draft:** Gerhard Blasche, Adrienne Chang, Otto Pichlhoefer.

**Writing – review & editing:** Gerhard Blasche, Jessica deBloom, Adrienne Chang, Otto Pichlhoefer.

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
