## [Decision Letter · Decision Letter 0]

24 Sep 2020

PONE-D-20-21236

Is a meditation retreat the better vacation? Effect of retreats and vacations on fatigue, emotional well-being, and mindfulness

PLOS ONE

Dear Dr. Blasche,

Thank you for submitting your manuscript to PLOS ONE. After careful consideration, we feel that it has merit but does not fully meet PLOS ONE’s publication criteria as it currently stands. Therefore, we invite you to submit a revised version of the manuscript that addresses the points raised during the review process.

We look forward to receiving your revised manuscript.

Kind regards,

Stefan Hoefer

Academic Editor

PLOS ONE

Journal Requirements:

Reviewers' comments:

Reviewer's Responses to Questions

**Comments to the Author**

1. Is the manuscript technically sound, and do the data support the conclusions?

Reviewer #1: Partly

Reviewer #2: Yes

2. Has the statistical analysis been performed appropriately and rigorously? 

Reviewer #1: I Don't Know

Reviewer #2: Yes

3. Have the authors made all data underlying the findings in their manuscript fully available?

Reviewer #1: Yes

Reviewer #2: Yes

4. Is the manuscript presented in an intelligible fashion and written in standard English?

Reviewer #1: No

Reviewer #2: Yes

5. Review Comments to the Author

Reviewer #1: Thank you for considering me to review this manuscript (“Is a meditation retreat the better vacation? Effect of retreats and vacations on fatigue, emotional wellbeing, and mindfulness”). As detailed in the manuscript, there is a growth of Buddhist Psychology in different parts of the world. Some of the Buddhist teachings have increasingly been employed to help individuals and organizations to cope with stress and distress. Along such the development, ashram, or, in modern parlance, retreats have cropped up to provide cloistered environments for the ‘seekers’ to get the breadth and depth of Buddhist therapy or ‘trio of life’. Previous studies have explored the functioning (psychological and emotional) of seekers during the entrance and upon ‘discharge’ or completion of the retreat. The questions arise on the durability or persistence of what has been gained from the retreat. This is an enticing empirical question which this manuscript has attempted to address. To fulfil the research objective, the authors did what it seems like an intervention study. The study has accrued three types of participants: (1) ‘MeR’ group (MEDITATION RETREATS), (2) ‘VwM’ group (VACATIONS WITH MEDITATION) and (3) ‘VoM’ group (VACATIONS WITHOUT MEDITATION). The outcome measures included Acting with awareness/ mindfulness that was tapped by the Kentucky Inventory of Mindfulness Skills, Fatigue solicited by Fatigue Assessment Scale) and wellbeing tapped by W HO-5 Well-Being Index. Since this study aimed to tease the durability of what was gained during the retreat, participants underwent three assessment occasions, one at Baseline (T1), 10 days (T2), and 10 weeks (T3). The authors observed the typological changes among the accrued sample (n=120) in terms of operationalized outcome measures- mindfulness, fatigue, and wellbeing. This study has the potential to shed light on the relative long-term benefit of Buddhist philosophy and technique. However, the authors were not diligent on the study design and approach to science. For the authors’ consideration, I will now focus on the ‘bigger pictures’ and if the authors would be able to rebut or circumvent the below-mentioned constraints, then the manuscript will deserve a full-fledged review for which I would be happy to contribute.

MAJOR ISSUES

1.STRUCTURE

The structure of the manuscript could be improved. For example, instead of using myriad subheadings. The authors could focus on the following: Introduction/Background, Method, Result, Discussion, Conclusion, and reference. The aims of the study should be clearly stated at the end of the introduction.

2. SAMPLE

The study failed to show the homogeneity of the sample. Some participants have had the “average of 10 years of meditation practice”. This represents an important confounder. The authors did not inform the reader how the sample size was calculated so that generalization of this study could be considered.

2. CONTEXT

To me, this study has all the veneers of the intervention study. According to the best practice, all intervention studies should be registered. The authors stated (approved by the” institutional review 184 board of Miami University, Oxford, OH, USA 45056 with the project reference number: 01554e:). Something is missing here. Please check.

3.CONCEPTUALIZATION

Since some of the participants were regular in meditation, conceptualize this study is not equipped to examine ‘durability’ or ‘recovery’. To match the authors’ aims, a different methodology will be required. This study, as alluded above, has features of the intervention study. Since the authors did not assign into three interventions, the study could be conceived as a ‘naturalistic observation’ or ‘preliminary study’.

4.LANGUAGE AND ABBREVIATION

As a non-native speaker, I could not escape noticing many problems in syntax and grammar as well as usage of bombastic words (“meditation training during vacation could alleviate the greatest pitfall of vacations”). Too many abbreviations have been thrown in the text. Their judicial use will be paramount to avoid confusing the reader.

Reviewer #2: This manuscript aims to address timely research questions on the effects of intensive meditation practice, answers to which are usually limited by an absence of meaningful control groups. A clear merit is the novelty and elegance of the authors’ methodological approach. This study has, in principle, the potential to offer an important contribution to the ongoing debate on the efficacy of intensive meditation practice. However, there are some important conceptual points that need to be addressed before this manuscript could be considered for publication. Please regard the suggestions below as an attempt to improve the positive impact of this study on this nascent research field. Please find the Comments to the Author attached. Thank you.

6. PLOS authors have the option to publish the peer review history of their article (what does this mean?). If published, this will include your full peer review and any attached files.

Reviewer #1: **Yes: **Samir Al-Adawi

Reviewer #2: No

---

## [Author Response · Author response to Decision Letter 0]

17 Nov 2020

Responses to reviewers

Reviewer #1

This study has the potential to shed light on the relative long-term benefit of Buddhist philosophy and technique. However, the authors were not diligent on the study design and approach to science. For the authors’ consideration, I will now focus on the ‘bigger pictures’ and if the authors would be able to rebut or circumvent the below-mentioned constraints, then the manuscript will deserve a full-fledged review for which I would be happy to contribute.

1. Structure: The structure of the manuscript could be improved. For example, instead of using myriad subheadings. The authors could focus on the following: Introduction/Background, Method, Result, Discussion, Conclusion, and reference. The aims of the study should be clearly stated at the end of the introduction.

Response: 

Thank you for carefully reading our manuscript. As suggested, we have deleted the subheadings from the introduction. We left the subheadings in the methods and results section for easier readability. We use the other main headings as suggested, i.e., Introduction, Method, Result, Discussion, Conclusion, and References. We now state the aims of the study in the second to last paragraph of the introduction, which also improves the structure in our view.

2. Sample: The study failed to show the homogeneity of the sample. Some participants have had the “average of 10 years of meditation practice”. This represents an important confounder. The authors did not inform the reader how the sample size was calculated so that generalization of this study could be considered.

Response: 

We have tested various differences between the three groups (e.g., background variables, baseline values of variables) and display results of the tests in Table 1. There were a few significant differences between the groups (i.e., partner, language, meditation practice per week during stay, relaxation, mastery, control). We have added all these potential confounders as control variables in the subsequent analyses. 

This study indeed concerns a naturalistic observation study and thanks to the reviewer for suggesting this clarification. Participants were not randomly assigned to experimental groups but freely chose whether they went on a vacation, whether they meditated on vacation or whether they chose to go on a meditation retreat. As the sample was recruited via a Buddhist organization, most participants identify themselves as Buddhist and have affinity and experience with meditation practices. 

We discuss this as a limitation of our study in the discussion section (last paragraph). We have also added a clear description of the study design, using the term “naturalistic observation” in the introduction, method section and in the abstract. 

3. Context: To me, this study has all the veneers of the intervention study. According to the best practice, all intervention studies should be registered. The authors stated (approved by the” institutional review 184 board of Miami University, Oxford, OH, USA 45056 with the project reference number: 01554e. Something is missing here. Please check.

Response: 

As stated above and suggested by the reviewer, this study is not an intervention study in which participants were randomized into experimental groups. Therefore, the study was not registered, but ethical review was requested and granted. We are clearer about our design in the method section now and refer to this study as a “naturalistic observation” (introduction, next to last paragraph; method section, design, first paragraph). We also checked the reference number of the report of the institutional review board which is correct.

4. Conceptualization: Since some of the participants were regular in meditation, conceptualize this study is not equipped to examine ‘durability’ or ‘recovery’. To match the authors’ aims, a different methodology will be required. This study, as alluded above, has features of the intervention study. Since the authors did not assign into three interventions, the study could be conceived as a ‘naturalistic observation’ or ‘preliminary study’.

Response: 

Please see our comments above regarding the study design we use. Most of the participants in our study are indeed experienced meditators and regularly practiced meditation for an average of 5 hours per week (see table 1). The aim of the present study was to test effects of meditation during a leisure episode (vacation, retreat), assuming that meditation will show a lasting improvement on well-being connected to improvements in mindfulness. However, we thank the reviewer for pointing out a potential limitation of the study: the effects may be due to changes in the extent of home-based meditation practice. We added this point as a potential limitation in the limitation section of our study (discussion, last paragraph). 

We have also changed the wording in the abstract and write now: “The aim of the present study was to investigate whether vacation effects would be more lasting if individuals practiced meditation during the leisure episode. Meditation is known to improve well-being in a durable fashion among others by enhancing the mental faculty of mindfulness. In this aim, leisure vacations during which individuals practiced meditation to some extent were compared with holidays not including any formal meditation practice as well as with meditation retreats (characterized by intense meditation practice) utilizing a naturalistic observational design.” 

5. Language and abbreviation: As a non-native speaker, I could not escape noticing many problems in syntax and grammar as well as usage of bombastic words (“meditation training during vacation could alleviate the greatest pitfall of vacations”). Too many abbreviations have been thrown in the text. Their judicial use will be paramount to avoid confusing the reader.

Response: 

Thank you for pointing out these weaknesses in our writing style. We have deleted the abbreviations from the text and tables and use the exact and complete terms now. This indeed improves readability. We have also removed the first paragraph in the introduction, including the “bombastic” wording. We carefully reread the manuscript and made sure that syntax and grammar are correct. Several minor changes were made to the text in this process. Also, one of our co-authors is a native speaker.

Reviewer #2

This manuscript aims to address timely research questions on the effects of intensive meditation practice, answers to which are usually limited by an absence of meaningful control groups. A clear merit is the novelty and elegance of the authors’ methodological approach. This study has, in principle, the potential to offer an important contribution to the ongoing debate on the efficacy of intensive meditation practice. However, there are some important conceptual points that need to be addressed before this manuscript could be considered for publication. Please regard the suggestions below as an attempt to improve the positive impact of this study on this nascent research field. 

Major point:

Overall, the description and discussion of meditation practice is too general and broad. A more careful presentation of the complexities and nuances of contemplative research is needed, including a discussion of theoretical frameworks of meditation practices through which the type of meditation practiced on these retreats can be conceptualised. Specific suggestions are offered below.

Response: 

Thank you for the positive feedback on our manuscript, careful reading and suggesting many helpful references to the literature that we had missed. We have done our best to incorporate references to these papers and improve our theoretical and conceptual framework. 

1. Introduction: The terms “mindfulness meditation” and “meditation” are often used interchangeably. In the introduction, please offer a more nuanced definition of meditation as a family of diverse practices that include but are not limited to mindfulness meditation. Please refer to landmark papers including:

• Dahl CJ, Lutz A, Davidson RJ. Reconstructing and deconstructing the self: cognitive mechanisms in meditation practice. Trends in cognitive sciences. 2015 Sep 1;19(9):515-23.

• Lutz A, Slagter HA, Dunne JD, Davidson RJ. Attention regulation and monitoring in meditation. Trends in cognitive sciences. 2008 Apr 1;12(4):163-9.

• Van Dam NT, Van Vugt MK, Vago DR, Schmalzl L, Saron CD, Olendzki A, Meissner T, Lazar SW, Kerr CE, Gorchov J, Fox KC. Mind the hype: A critical evaluation and prescriptive agenda for research on mindfulness and meditation. Perspectives on psychological science. 2018 Jan;13(1):36-61.

Response: 

Thank you for pointing out this weakness in our definition. In general, the manuscript has been edited to use a more nuanced definition of meditation, with reference to published conceptual framework/ taxonomy offered from key papers as suggested by reviewer. Specifically, we have adapted the definition/description as suggested and also included reference to the three papers mentioned above. We write now: (introduction, 3rd paragraph): “Meditation can be understood as a family of diverse practices that include, but are not limited to, mindfulness meditation. Found within Buddhism and other contemplative traditions, such meditation practices seek to cultivate and regulate specific psychological processes lastly aimed at increasing well-being. Recently, these practices have been grouped into three families, the attentional, the constructive and the deconstructive family {Dahl, 2015 #5328}. These families not only include different practices but also affect well-being through different avenues. The attentional family for example, encompassing mindfulness meditation and mindfulness based stress reduction, improves well-being by a process known as “meta-awareness” or “cognitive distancing”, i.e. “stepping back and observing one’s internal processes of thinking and feeling” {Dahl, 2015 #5328}. This includes overcoming experiential fusion, that is being absorbed in the contents of consciousness, which reduces our ability to monitor and/or regulate psychological processes. To be able to promote these processes, two cognitive faculties have to be developed: focused attention, i.e. “sustaining selective attention moment by moment on a chosen object” and open monitoring, i.e. “nonreactively monitoring the content of experience” {Lutz, 2008 #5329}”.

2. It is unclear why the manuscript starts off with a general statement about books published on “mindfulness”. Please consider deleting the first sentence of the introduction.

Response: 

We have deleted the first sentence of the introduction as suggested. Due to a more thorough revision of the introduction, we deleted the entire first paragraph. We used the saved space to instead provide a more detailed definition and description of meditation practices (see our response above). 

3. It is not clear why an elaborate presentation of research on mindfulness-based interventions is needed in the context of the present manuscript. On several levels, intensive meditation retreats differ markedly from standard mindfulness-based programmes. Please consider embedding the narrative of this manuscript within previous research on meditation retreats. 

Response: 

We agree with the reviewer that a shifting of the narrative towards retreats mirrors the aims of the present study more closely and thus would improve the manuscript. Therefore, we expanded our discussion on meditation retreats, adding two new paragraphs (introduction, paragraph 5 & 7) and explicitly acknowledge the differences between retreats and standard mindfulness-based programmes in our description of meditation retreats (introduction, 5th paragraph). We kept a short summary of effects of standardized mindfulness programs (introduction, 4th paragraph) to illustrate the effects of mindfulness meditation, a form of meditation both used during retreats as well as by some of the study participants during vacation, constituting one of our study groups. 

4. Please add a discussion of the systematic reviews and theoretical papers listed below. Please also use them to elaborate on the limitations and future directions of research on meditation retreats.

• McClintock AS, Rodriguez MA, Zerubavel N. The effects of mindfulness retreats on the psychological health of non-clinical adults: A meta-analysis. Mindfulness. 2019 Aug 15;10(8):1443-54.

• Khoury B, Knäuper B, Schlosser M, Carrière K, Chiesa A. Effectiveness of traditional meditation retreats: A systematic review and meta-analysis. Journal of Psychosomatic Research. 2017 Jan 1;92:16-25.

• King BG, Conklin QA, Zanesco AP, Saron CD. Residential meditation retreats: their role in contemplative practice and significance for psychological research. Current opinion in psychology. 2019 Aug 1;28:238-44.

Response: 

We thank the reviewer for suggesting these relevant publications. In the revised version of the manuscript, we summarize the results of the meta-analyses (introduction, 7th paragraph) and included insights of King et al. (2019) in our description of retreats (introduction, 5th paragraph). In addition, we have included these references in our discussion of limitations as well as in the section on recommendation for future research (conclusions). 

5. “Propelled by research and relatively distanced from its spiritual roots, mindfulness radiated through medicine into society at large.” Please consider reframing this. It seems that the retreats included in this study took place within an explicitly Buddhist framework. 

Response: 

We have removed the entire paragraph, including the problematic sentence (see also response to point 2). The retreats included in this study did indeed take place within a Buddhist framework as also stated in the methods section. Thank you for pointing this out. 

6. Page 5, line 113: “While a meditation retreat aims at developing non-judgmental present moment awareness through meditation practice, …”. This is too broad and general. What types of meditation were primarily practiced during these particular retreats (see comment 4 below)? Please describe why you have chosen retreats that have focused primarily on “mindful-awareness”. Also, the term “mindful-awareness” is not commonly used in meditation research. Please clarify and specify the terminology in accordance with published conceptual frameworks (e.g. Dahl et al. 2015). 

Response: 

We have removed this brief description and included a more precise description of the primary meditation practiced during the retreats elsewhere (introduction, 6th paragraph) referring to the suggested conceptual frameworks. In the chosen retreats, the primary form of meditation taught was Shamata meditation based on the Buddhist background of the organization. 

7. Methods: Page 7, line 213: “The formal sitting practice of meditation taught at these retreats is mindful-awareness, where the practitioner learns to focus their attention on an object of meditation, often the breath, to cultivate awareness of one’s inner world: thoughts, emotions, sensations, perceptions.” Please elaborate on the type of practice that was used, its perceived goals, and the cognitive mechanism it primarily cultivates. Please try to embed this type of practice within a theoretical taxonomy of meditation practices (e.g. Dahl et al. 2015) to differentiate it from other practices.

• Reference: Dahl CJ, Lutz A, Davidson RJ. Reconstructing and deconstructing the self: cognitive mechanisms in meditation practice. Trends in cognitive sciences. 2015 Sep 1;19(9):515-23.

Response: 

We thank the reviewer for demanding clarity in regard to the meditation practices used during the retreats. In the revised version of the paper, we describe the types of meditation practiced during the retreats and embed the practices with the proposed theoretical framework. We also included the discussion of the meditation practices, their theoretical underpinning and their cognitive mechanisms in the introduction (paragraph 4 & 5). In the method section, we state the major forms of meditation practice engaged in during retreats in accordance with the proposed theoretical framework to clarify the practices used. 

8. Could you please include (either in methods section or as supplementary material) the information that was given to participants during recruitment, including the wording of the aims of the study. Could there have been a likelihood that participants believed that the study might aim to demonstrate that meditation retreats (or vacation with meditation) are superior to vacations? If so, please discuss this as a limitation.

Response: 

Behavioral interventions require participants to be aware on the purpose of the study in order to adhere to the behavioral guidelines. So, motivation and honest information on the purpose of the study is not only ethically required, but also necessary in order to ensure that the study can be effective. As suggested, we have added the text of the recruitment e-mails to the supplementary materials. After reviewing the invitation letters again, we do not believe that we induced the belief that retreats were superior to vacations in any way. Thus, we did not add this point to the limitation section. 

9. Please specify the types of meditation practices that participants in the vacation-with-meditation group engaged in. If those differed from the types of meditation practiced on retreat, then please highlight and discuss this when interpreting the results.

Response: 

Unfortunately, we did not assess the type of meditation practices that participants in the vacation-with-meditation group engaged in. However, it is likely that this individual meditation closely resembled the practices engaged in during the retreats, i.e. Shamata and Tonglen, as all individuals were members of the same Buddhist organization. We added this information to the method section, description of retreat- and vacation stays.

10. Results: If available, could you please add to Table 1 the number (%) of participants in each group who have previously been on retreat. 

Response: 

Regretfully, this information is not available and can thus not be added to the table. 

11. Discussion: To nuance the discussion, please briefly mention potential downsides of meditation retreats that are unlikely to be encountered on vacation without intensive meditation practice. For instance, a recent meta-analysis highlights potential meditation-related challenges (Farias et al., 2020) and a large study of regular meditators indicated that individuals with retreat experience were more likely to report unpleasant meditation-related experiences (Schlosser et al., 2019). 

• Farias M, Maraldi E, Wallenkampf KC, Lucchetti G. Adverse events in meditation practices and meditation‐based therapies: a systematic review. Acta Psychiatrica Scandinavica. 2020 Aug 7.

• Schlosser M, Sparby T, Vörös S, Jones R, Marchant NL. Unpleasant meditation-related experiences in regular meditators: Prevalence, predictors, and conceptual considerations. PloS one. 2019 May 9;14(5):e0216643.

Response: 

Thank you for pointing out these issues. We have gladly included reference to these papers and the potential downsides of retreats. We have added a paragraph discussing these potential adverse effects of meditation (Discussion, 8th paragraph). 

12. Tables: Please add full names of MeR, VvM, VoM, AWA etc. to the footnote of each table. 

Response: 

In line with point 5 of reviewer 1, we deleted all abbreviations related to variables from the manuscript. In addition, we included the full names of remaining abbreviations in the footnote of the tables. Thank you for pointing this out. This measure has indeed improved readability. 

We would like to thank the reviewers for their feedback and hope that they also feel that our paper improved a lot thanks to their constructive comments.

---

## [Decision Letter · Decision Letter 1]

11 Dec 2020

PONE-D-20-21236R1

Is a meditation retreat the better vacation? Effect of retreats and vacations on fatigue, emotional well-being, and mindfulness

PLOS ONE

Dear Dr. Blasche,

Thank you for submitting your manuscript to PLOS ONE. After careful consideration, we feel that it has merit but does not fully meet PLOS ONE’s publication criteria as it currently stands. Therefore, we invite you to submit a revised version of the manuscript that addresses the points raised during the review process.

I kindly ask you to address reviewers 2 comments carefully. As reviewer 2 states, after consideration of these remaining minor issues, acceptance of the manuscript can be recommended. 

We look forward to receiving your revised manuscript.

Kind regards,

Stefan Hoefer

Academic Editor

PLOS ONE

Reviewers' comments:

Reviewer's Responses to Questions

**Comments to the Author**

1. If the authors have adequately addressed your comments raised in a previous round of review and you feel that this manuscript is now acceptable for publication, you may indicate that here to bypass the “Comments to the Author” section, enter your conflict of interest statement in the “Confidential to Editor” section, and submit your "Accept" recommendation.

Reviewer #1: All comments have been addressed

Reviewer #2: (No Response)

2. Is the manuscript technically sound, and do the data support the conclusions?

Reviewer #1: Yes

Reviewer #2: Yes

3. Has the statistical analysis been performed appropriately and rigorously? 

Reviewer #1: Yes

Reviewer #2: Yes

4. Have the authors made all data underlying the findings in their manuscript fully available?

Reviewer #1: Yes

Reviewer #2: Yes

5. Is the manuscript presented in an intelligible fashion and written in standard English?

Reviewer #1: Yes

Reviewer #2: Yes

6. Review Comments to the Author

Reviewer #1: Thank you for considering me to review this manuscript, “Is a meditation retreat the better vacation? Effect of retreats and vacations on fatigue, emotional well-being, and mindfulness”. I have read with interest in the revised manuscript. I have paid particular attention to two reviewers’ comments, my counterpart, and myself. I have raised issues pertinent to the methodological approach while my counterpart has commented on the conceptualization issue of the study. The authors appeared to have responded or rebutted the protracted comments from two reviewers. Otherwise, the authors have highlighted some of the un-addressable points as the limitations of the study. In my opinion, the scientific merit of the manuscript has significantly improved. On this ground, I have no hesitation to recommend this manuscript for publication.

Reviewer #2: I thank and commend the authors for diligently addressing my conceptual concerns. I believe that the impact of the paper and its resilience to critics has been substantially improved. Importantly, the authors have added a more nuanced discussion of the nascency and heterogeneity still characterising meditation research. After addressing the minor comments below, I would recommend this manuscript for publication.

Line 113

“Thus, within a retreat, meditation practice invites a practitioner to be more open and non-judgmental to one’s experience, without labeling them as pleasing or unpleasing, through repetitive familiarization with one’s cognitive processes as well as the content of one’s consciousness. As such, meditation is geared at “seeing things as they are “.

Could you please delete or substantially reframe this section? Indeed, some meditative traditions and schools purport that particular meditative practices (e.g. Goenka vipassana retreats, certain forms of mindfulness training) help us to see “things as they are”. Other frameworks, however, criticise this approach because it rests on a set of unquestioned assumptions (e.g. “there really is a real reality a meditator can see”) and prevents a deeper inquiry into the dependent arising and empty nature of all phenomena (e.g. see Burbea, 2014). Further, some meditative practices (e.g. samadhi practice, forms of samatha practice) actually encourage meditators to cultivate profound states of well-being (e.g. jhanas) by actively inclining the mind towards pleasant sensations and abiding in them, thereby actively re-habituating the mind to not get drawn into the difficult and unpleasant perceptions. Some meditative practices are very passive and receptive in their approach; others are very proactive; different practices will be helpful for different people at different times. Again, all this speaks to the rich diversity of meditative practices and implies that using the term meditation in a broad and generalised manner (“meditation is geared at …”) is likely to add more confusion than clarity to this nascent research field. When re-reading your manuscript to make final edits, please also consider other sections in your manuscript where the (very understandable) tendency to generalise might be apparent.

References:

Burbea, R. (2014). Seeing that frees: Meditations on emptiness and dependent

arising. Hermes Amāra Publications.

Line 147

“ones meditation practice”

Please correct: “ones” to “one’s”

Line 168

“… by various psychological mechanisms associated with an increase in mindfulness.”

Please change “associated with” to “including”. Otherwise, it can sound as if mindfulness is the primary cognitive mechanism by which all meditative practices exert their effects.

Discussion

Could you please add 1-2 sentences briefly noting that other psychological processes not assessed in this study might also play an important role in improving well-being and reducing suffering in the context of a regular meditation practice. These potential mediators include self-compassion (e.g., Baer et al., 2012, n = 77 meditators; Schlosser et al., 2020, n = 1281 meditators), neuroticism, and perceived control (Jacobs et al., 2011; already cited in your manuscript). You could recommend that future research on the effects of vacation, meditation retreats, and regular meditation practice consider a broader range of potential mediators.

References:

Baer, R.A., Lykins, E. L. B.,& Peters, J. R. (2012). Mindfulness and self-compassion as predictors of psychological wellbeing in long-term meditators and matched non-meditators. The Journal of Positive Psychology, 7(3), 230–238. https://doi.org/10.1080/17439760.2012.674548

Jacobs, T. L., Epel, E. S., Lin, J., Blackburn, E. H., Wolkowitz, O. M., Bridwell, D. A., ... & King, B. G. (2011). Intensive meditation training, immune cell telomerase activity, and psychological mediators. Psychoneuroendocrinology, 36(5), 664-681.

Schlosser, M., Jones, R., Demnitz-King, H., & Marchant, N. L. (2020). Meditation experience is associated with lower levels of repetitive negative thinking: the key role of self-compassion. Current Psychology. https://doi.org/10.1007/s12144-020-00839-5

Discussion

Could you please add one sentence that your results might be more a reflection of the short 5-item well-being measure you utilised rather than the latent well-being construct it was intended to capture. You could recommend that future research explores whether your findings can be replicated with longer and more comprehensive well-being measures.

7. PLOS authors have the option to publish the peer review history of their article (what does this mean?). If published, this will include your full peer review and any attached files.

Reviewer #1: **Yes: **Samir Al-Adawi

Reviewer #2: No

---

## [Author Response · Author response to Decision Letter 1]

12 Jan 2021

Responses to reviewers

We would like to thank the reviewer and the editor for the renewed review of our manuscript and for the positive assessment. Below, we address the remaining issues raised by Reviewer 2 point by point. 

We are looking forward to receiving your feedback on these changes. 

Reviewer #2

Points:

1. Line 113: “Thus, within a retreat, meditation practice invites a practitioner to be more open and non-judgmental to one’s experience, without labeling them as pleasing or unpleasing, through repetitive familiarization with one’s cognitive processes as well as the content of one’s consciousness. As such, meditation is geared at “seeing things as they are “.

We deleted this section as suggested. We understand that this statement may have been unsatisfactory in various ways and did not reflect the diversity of meditation practices sufficiently. 

2. Line 147: corrected

3. Line 168: “… by various psychological mechanisms associated with an increase in mindfulness.” We changed “associated with” to “including” as suggested.

4. Discussion, end of 4th paragraph: We thank the reviewer for this suggestion and added the following paragraph: “It should be noted that other psychological processes associated with meditation practice not included in this study might also play a role in improving well-being, such as self-compassion [60,61], neuroticism, and perceived control [31]. We suggest that future research on vacation, meditation retreats, and regular meditation practice should consider a broader range of potential mediators.” We have also included the suggested literature (Baer, 2012; Schlosser, 2020). 

5. Following the advice to add one sentence regarding the relatively short well-being measures we used, we added the following paragraph in the discussion: “Fourthly, we used a relatively short scale (i.e., 5 items) to assess emotional well-being. Even though this scale has good psychometric qualities demonstrating high levels of reliability, validity as well as sensitivity [66], future research could replicate our findings with more comprehensive scales and broaden the well-being construct to include eudaimonic well-being, thriving and meaning [67,68].” In addition, the results for emotional well-being and fatigue are comparable in our study, thus supporting the validity of the WHO-5 measure.

---

## [Editor Report · Decision Letter 2]

13 Jan 2021

Is a meditation retreat the better vacation? Effect of retreats and vacations on fatigue, emotional well-being, and acting with awareness

PONE-D-20-21236R2

Dear Dr. Blasche,

We’re pleased to inform you that your manuscript has been judged scientifically suitable for publication and will be formally accepted for publication once it meets all outstanding technical requirements.

Kind regards,

Stefan Hoefer

Academic Editor

PLOS ONE
---

## [Editor Report · Acceptance letter]

21 Jan 2021

PONE-D-20-21236R2 

Is a meditation retreat the better vacation? Effect of retreats and vacations on fatigue, emotional well-being, and acting with awareness 

Dear Dr. Blasche:

I'm pleased to inform you that your manuscript has been deemed suitable for publication in PLOS ONE. Congratulations! Your manuscript is now with our production department. 

Kind regards, 

on behalf of

Dr. Stefan Hoefer 

Academic Editor

PLOS ONE